# MME-YOLO: Multi-Sensor Multi-Level Enhanced YOLO for Robust Vehicle Detection in Traffic Surveillance

**DOI:** 10.3390/s21010027

**Published:** 2020-12-23

**Authors:** Jianxiao Zhu, Xu Li, Peng Jin, Qimin Xu, Zhengliang Sun, Xiang Song

**Affiliations:** 1School of Instrument Science and Engineering, Southeast University, Nanjing 210096, China; jx_zhu@seu.edu.cn (J.Z.); 220173203@seu.edu.cn (P.J.); Jimmy.xqm@gmail.com (Q.X.); 2Traffic Management Research Institute, Ministry of Public Security, Wuxi 214151, China; SZL8205@sina.com; 3School of Electronic Engineering, Nanjing Xiaozhuang University, Nanjing 211171, China; sx2190105@163.com

**Keywords:** vehicle detection, multi-sensor fusion, complex scenes, multi-scales, smart city

## Abstract

As an effective means of solving collision problems caused by the limited perspective on board, the cooperative roadside system is gaining popularity. To improve the vehicle detection abilities in such online safety systems, in this paper, we propose a novel multi-sensor multi-level enhanced convolutional network model, called multi-sensor multi-level enhanced convolutional network architecture (MME-YOLO), with consideration of hybrid realistic scene of scales, illumination, and occlusion. MME-YOLO consists of two tightly coupled structures, i.e., the enhanced inference head and the LiDAR-Image composite module. More specifically, the enhanced inference head preliminarily equips the network with stronger inference abilities for redundant visual cues by attention-guided feature selection blocks and anchor-based/anchor-free ensemble head. Furthermore, the LiDAR-Image composite module cascades the multi-level feature maps from the LiDAR subnet to the image subnet, which strengthens the generalization of the detector in complex scenarios. Compared with YOLOv3, the enhanced inference head achieves a 5.83% and 4.88% mAP improvement on visual dataset LVSH and UA-DETRAC, respectively. Integrated with the composite module, the overall architecture gains 91.63% mAP in the collected Road-side Dataset. Experiments show that even under the abnormal lightings and the inconsistent scales at evening rush hours, the proposed MME-YOLO maintains reliable recognition accuracy and robust detection performance.

## 1. Introduction

Vehicle detection is a well-known question in traffic scenes, especially for intelligent vehicles. With such algorithms, smart cars can identify and predict dynamic targets in the surrounding environment, thereby reducing accidents such as collisions. However, a series of serious traffic accidents show that detection algorithms running on-board is not reliable enough as only limited perspectives are used, such as the accident of the Tesla Model S in 2016 and the accident of Uber automobile in 2014. Essentially, neither line-controlled smart cars nor rich-experienced drivers can swiftly react to the suddenly appeared targets with limited perspectives.

Therefore, it is natural to consider measuring by the cooperative roadside systems in traffic surveillance in areas with frequent accidents. Through the vehicle detection and behavior prediction algorithm, the cooperative roadside system can give early warning signals to vehicles that may have accidents in the near future. Therefore, these accidents can be sharply reduced.

Unfortunately, there is a huge technical gap between these cooperative roadside systems and the conventional traffic surveillance devices. As illustrated in Figure 1, the conventional traffic surveillance devices are served for traffic flow analyses and traffic management, which concerns only about the near easily-recognized objects. However, for the cooperative roadside system, its concentrates on the movement of the target in the field of view. Thus, both large-scale and small-scale vehicles are required to be captured correctly. At the same time, as a safety warning system, the robust detection ability is even more addressed under realistic adverse conditions, such as the typical occlusion and the backlights at night. Generally, traditional vehicle detection algorithms are hard to meet the requirement of such an application. For filling these gaps, a series of deep-learning-based methods have been adopted in traffic surveillance devices. This kind of data-driven method achieves pretty good results by simply re-training fantastical object detection algorithms.

However, the natural scales problem in traffic surveillance often troubles these visual detectors. Compared with the small-scale vehicles, the large-scale objects receive more supervised information in visual feature maps. Thus, the gradient of the small-scales is easily overwhelmed by that of large-scale ones. Consequently, the accuracy of the small-scale vehicles is often much lower than the large-scales. Another realistic question is that the visual discriminative representations between frontal vehicles and the background are generally disturbed by the change of ambient light. Besides, the occlusion is often combined with such abnormal lighting scenarios, such as the rush hours at nightfall.

Under this setup, detection algorithms based on the color images are hard to distinguish vehicle targets among the confusing background. Thus, it is reasonable to combine multiple types of data to get a reliable outcome. In the field of intelligent transportation, sensors widely used are the camera, millimeter-wave radar, and LiDAR. The infrared camera is a special camera working well in weak illuminations. However, it struggles to separate the objects in the congested scenes when only the single-channel temperature information is adopted. The neuromorphic camera is also an explosively developed product that captures the boundaries of the objects. However, its limited spatial resolution makes it hard to recognize the small-scale objects. Looking back at the active ranging sensors, compared with the millimeter-wave radar, LiDAR has higher resolution and reliable three-dimensional outputs. Therefore, LiDAR is more accessible when using data-dependent deep networks. Moreover, as an active sensor instead of a passive receiver, LiDAR keeps consistent accuracy towards variable lighting conditions and the occlusions as the distance information is presented. Considering both the requirements of neural networks and the characteristics of different sensors, the fusion of both the three-dimensional point clouds and the color images is respected as a good solution in the cooperative roadside system.

To alleviate the problems of low precision caused by the dramatic changes in the vehicle scales and illuminations along with occlusion, a multi-sensor multi-level enhanced convolutional network architecture (MME-YOLO) is proposed to composite the point clouds and color images for better inference abilities towards vehicle detection in traffic surveillance. Specifically, we first improve the feature fusion part of the single-stage object detector YOLOv3 [1] by introducing the attention-guided feature selection blocks. Besides, inspired by the anchor-free idea, we integrate the anchor-based/anchor-free inference heads and above attention blocks to form an enhanced visual network (E-YOLO). Progressively, the visual hue information, depth, and height information from point clouds are concatenated into a regular format to feed into the neural subnet of LiDAR. These feature maps in the LiDAR subnet are further added to the Image subnet at different semantical levels. With this special design, even under abnormal lighting conditions or congested conditions, the integrated MME-YOLO can classify the vehicle objects and regress their location accurately. Principally, the main contributions of this paper are as follows:To improve the performance of vehicle detection under realistic adverse conditions in traffic surveillance, we introduce an innovative multi-sensor robust detection structure, which collects the information of point clouds and color images at various levels. To the best of our knowledge, this paper is the first one using recursively-composite multi-sensor fusion strategies to obtain accurate detections.To promote the effectiveness of the network to deal with multi-scale vehicles in traffic surveillance, we propose attention-guided feature selection blocks for effective screening and merging different level information. Compared with several recent works, this structure can improve the detection accuracy of the multi-scale targets in hybrid complex scenes effectively.

The rest of this paper is organized as follows. Some related works in vehicle detection and multi-sensor fusion strategies are reviewed in Section 2. Then, we describe the proposed approach elaborately in Section 3. Related experiment results are presented and discussed in detail in Section 4. At last, we make the conclusions in Section 5.

## 2. Related Work

Vehicle detection has achieved remarkable success since the advance of deep-learning methods. Especially for the on-board algorithms, not only has the visual neural network been extensively studied, but also the multi-sensor fusion method has also been elaborated. However, due to the natural differences between observation points, the characteristics of detectors exhibit a huge gap between the traffic surveillance perspective and the onboard one. In this part, we first briefly review the related works of traditional vehicle detectors in the traffic surveillance system, then we introduce the deep-learning-based vehicle detectors from the vehicle perspective, then we compare several multi-sensor fusion architectures from on-board views, and finally, we introduce the attention skills used in visual detection networks for multi-scale objects.

### 2.1. Traditional Vehicle Detectors in Traffic Surveillance Systems

Vehicle detection is once regarded as a dynamic object detection problem. Three kinds of approaches are developed to solve this problem, including optical flow, background subtraction, and temporal difference. Optical flow is based on the assumption of consistent lighting intensity in the region of interests (ROIs), so it is very sensitive to the change of ambient illumination. As for the temporal difference, it detects dynamic objects by calculating the differences between consecutive frames. The shapes of detected objects are often broken, and the recall of detection is unacceptable in congested scenes such as cars are moving slowly in heavy traffic. When looking into the background subtraction, it receives wide concern since Robust Principle Component Analysis (RPCA) [2] is adopted for modeling the foreground and background in traffic surveillance. Gaussian Mixtures Models (GMM) [3] is also considered an accurate model for background pixels. However, these models often require a series of initialization procedures, such as calculating the input observation matrix through batches of videos.

PNNMD [4] firstly improve background modeling procedure through probabilistic neural networks. With the remarkable power in feature extraction, PNNMD achieves a huge improvement than all other state-of-art traditional methods in all listed scenarios, like an expressway, freeway, and urban roads. However, as convolutional networks show their excellent visual works, more representative backbone models are adopted in vehicle detection algorithms.

### 2.2. Deep-Leaning-Based Vehicle Detectors from the Vehicle Perspective

With the advance of large automobile datasets like KITTI [5], APOLLO [6], and BDD100K [7], the precision of vehicle detectors have gained huge improvement from the onboard perspective. However, due to the limitation of the self-driving perspective, the perceived targets often have serious occlusion and inconsistent scale problems. To alleviate both of the two problems, FPN [8] (the feature pyramid fusion network), which combines high-level semantic information with low-level visual information, is designed to reason objects on different scales. Although this method gains significant improvement compared with single-inference-layer ones, the gradient of small-scale vehicles is often overwhelmed by a large number of low-level cues in high-resolution feature maps, which leads to false negatives. PANet [9] adopts another bottom-up branch to make the underlying information uploaded in fewer residual nodes to enhance the semantic information of low-level feature maps. NAS-FPN [10] uses Neural Architecture Search (NAS) technology to generate an uninterpretable FPN structure with more excellent performance. In the ASFF [11] network, it improves this by performing weight learning on the information of different layers. The feature pyramid, in EfficientDet [12], after heuristically revising the PANet network, also combined with the idea of weight learning and a cascading structure enhancement.

The idea of these works could be adopted appropriately, as scale inconsistencies also or even more seriously existed in roadside scenes. However, both these recent enhanced works partially ignored the imbalance of visual information between the background and the objects in adjacent maps. Especially for the small-scale vehicles, a large number of redundant visual features have brought great difficulties to small-scale vehicle detection. In our approach, the attention guided feature selection blocks are proposed for filtering useless information and strengthening the meaningful ones. Moreover, we tentatively solve the multi-scale imbalance more closely by integrating the multi-sensor fusion backbone.

### 2.3. Deep-Leaning-Based Multi-Sensor Fusion Detectors from the Vehicle Perspective

Point clouds tend to be disordered and irregular. Therefore, many multi-sensor fusion neural networks firstly adopt projection methods to reshape the LiDAR data into formal image format and then use well-developed visual backbone networks to analyze the depth and reflection information. MV3D [13] firstly elaborates on both the bird view and the frontal view to regress locations of vehicles in three dimensions (3D) from the self-driving perspective. Then, LMNet [14] specially designs a network for the point clouds, which incorporates the dilated convolution and 3D object regression loss. F-PointNet [15] alternatively adopts a convolutional visual network to get coarse locations on image coordination, which combines with a frustum point cloud to refine its bounding box in a cascade fashion.

Since both these works only introduce LiDAR information as an additional input modality or cascading revision factors, the relation between LiDAR data and Image data at the semantic level is seriously neglected. For example, the height in points can be utilized as a natural position filter for filtering abundant background messages in visual feature maps, which acts like a semantical attention mask. In our work, instead of direct fusing from the input, we combine the visual cues with semantic-like point maps by transmitting the same level feature maps in the LiDAR subnet to the Image subnet. Our experiments show this design achieves better performance in the test-set than fusing from the input.

### 2.4. Attention Skills Used for Multi-Scale Objects

As analyzed above, detectors from the traffic surveillance perspective face challenges from imbalanced scales. Recently, attention mechanisms are introduced to alleviate the above question in traffic surveillance by transfer learning. The typical attention structure can be divided into two types. The first kind is explicit attention, such as SENet [16], Residual Attention [17], CBAM [18], SKNet [19], ResneSt [20], etc. Another is implicit attention, such as Maxpooling, Gate control, etc. The accuracy after integrated these attention mechanisms achieves satisfactory improvement under marginal parameter costs. Although these excellent methods have carefully studied the effective way to split the frontal objects and background, the shortage of a single modality could not be relieved for the reason of different lighting conditions in real scenes. In our approach, not only conventional attention strategies have been adopted, but also the multi-sensor enhanced feature maps are stacked to act like implicit attention mask for additional support.

To sum up, considering the characteristics carried by the roadside perspective and the defects of existing multi-sensor fusion schemes, our team put up a specifically designed multi-sensor multi-level composite-backbone structure to gather different level’s semantic information from different sensors to achieve robust vehicle detection.

## 3. The Proposed MME-YOLO Network

In this section, we give an intensive description of the MME-YOLO network, and then demonstrate our design from the following aspects: the visual enhancement of YOLOv3 for more accurate multi-layer inference; the generation of the inputs of LiDAR subnet; multi-sensor multi-level composite fusion structure for robust vehicle detection.

### 3.1. Overall Network Structure

The overall architecture of MME-YOLO is illustrated in Figure 2. This framework includes three parts. The left part is a two-subnet composite fusion module, which takes the benefits from LiDAR and Color Images’ multi-level semantical information. Inside both subnets, down-sampling and convolutional operations are adopted to transition the feature maps between three inference levels. After the addition of different sensors’ feature maps, two cross-level attention blocks are introduced to filter out tremendous useless features in high-resolution feature maps and strengthen meaningful features in low-resolution ones in the medium part. In the right part, inspired by the idea of Anchor-Free, the anchor-based/anchor-free ensemble inference head is built for consideration of better generalization abilities in real scenes. The attention blocks and the ensemble head are summarized as a visual enhancement in short.

### 3.2. E-YOLO: Visual Enhancing Structures for Accurate Multi-Scale Inference

The multi-layer fusion part of YOLOv3 upsample the low-resolution feature maps in the deep layer and concatenate the lower one with high-resolution feature maps. This kind of feature fusion structure improves the accuracy by brutally concatenating features, ignoring the relation between deep layers and shallow layers in two dimensions. Specifically, the low-level feature maps are occupied with useless background features, which conforms to an imbalanced positive-negative distribution. Then, although large-scale objects do little rely on surroundings, small-scale objects usually need global context value for recognition. Thus, it is natural to employ different strategies, rather than the unified concatenation operation, for different scales.

A cross-level attention plugin is proposed to alleviate the above problems in the feature pyramid by highlighting profitable values while penetrating the excess and ineffective information in a bidirectional way. There are two kinds of cross-level attention blocks in our work. As illustrated in Figure 3, the top-down block up-samples the high-level semantic values to highlight the meaningful position in high-resolution maps and the bottom-up block retains the local low-level cues for deeper layers.

The calculation formula for the attention block is as follows:(1)$$T\left(x,y,z\right)=\left(\frac{1}{1+e^{-\left(s_{2}\left(x,y,z\right)\right)}}+1\right)\;*\;S\left(x,y,z\right)$$

Our top-down branch is similar to the idea of the Residual-Attention [14] structure, but there are several key differences. First, our attention module adopts two different blocks for feature selection, as the top-down block and the bottom-up block, which shows the reliability in experiments. Second, the proposed attention mask is naturally based on the detectors’ multi-layer structure, thus huge parameter costs are avoided. Third, the feature maps before attention structure are already partly enhanced by semantic-like LiDAR data. It strongly relieved the pressure of the attention mask for recognizing the small-scale objects.

In addition to the relation between layers, we also notice the generalization of the algorithm in the real scene. We propose an ensemble inference head by combining both the anchor-free branch and the anchor-based one which is adopted from the FSAF [21] structure. The traditional anchor-based branch is trained by allocating the positive label to anchors, which maximizes the interaction over unions (IOUs) between default anchors and ground truth labels. The loss of the anchor-based branch $loss_{ab}$ is calculated by the balanced summary over samples and levels. Specifically, $loss_{focal}^{l}$ refers to the focal loss [22] over vehicles’ category and $loss_{DIOU}^{l}$ refers to the DIOU [23] loss of vehicles’ bounding box on the $l_{th}$ level. On the contrary, the gradient of the anchor-free branch is calculated through automatic searching the layer of minimum loss $l^{*}$ from all inference layers. Then, this branch adopts the optimal level $l^{*}$ to calculate the loss of the anchor-free routine $loss_{af}$ by summarizing all samples in one batch. Finally, the overall loss of ensemble system $loss_{en}$ is obtained by weighted addition between the anchor-based loss and the anchor-free one. The balanced ratio parameters α, β, and θ are respectively chosen as 0.8, 0.4, and 0.4 by experiments on validation datasets. The overall processes are summarized as following:(2)$$loss_{ab}=\underset{n}{\sum }\underset{l}{\sum }\left(\theta \;*\;loss_{focal}^{l}+\left(1-\theta \right)\;*\;loss_{DIOU}^{l}\right)$$
(3)$$l^{*}=\underset{l}{\mathrm{argmin}}\left(\beta \;*\;loss_{focal}^{l}+\left(1-\beta \right)\;*\;loss_{DIOU}^{l}\right)$$
(4)$$loss_{af}=\underset{n}{\sum }\left(\beta \;*\;loss_{focal}^{l^{*}}+\left(1-\beta \right)\;*\;loss_{DIOU}^{l^{*}}\right)$$
(5)$$loss_{en}=\alpha *loss_{af}+\left(1-\alpha \right)\;*\;loss_{ab}$$

After integrating the anchor-based and anchor-free structure, the inference head could relatively alleviate the miss allocation problem of nearby targets, which happens frequently in the anchor-free module. This special design significantly improves the recall of multi-scales vehicles in congested scenes.

Although the above visual enhanced network gets benefits from the dedicated design, the hybrid complex scenarios bring great challenges to the robustness of visual detectors, which can hardly be solved by a single input modality. In the next part, we will analyze the characteristics of LiDAR data from the roadside perspective and describe the effective way of introducing point information to the visual backbone network.

### 3.3. LiDAR Point Image Generation

Different from the passive camera, LiDAR measures environmental information by emitting a laser beam and receiving the light wave signal that it returns. Thus, LiDAR has information on distance and depth that is missing from visual sensors. This special information modality has a unique advantage for identifying occlusions. Besides, LiDAR achieves reliable performance in all daytime as the narrow beam technology is integrated. The shortcomings of this sensor are also obvious. For the reason of the sparse resolution, it is difficult to conduct the fine classifications on the points. For example, Vans and Cars are hard to be separated. Moreover, for its irregularity, the sparse point clouds cannot be directly used as input to the regular deep learning methods.

Several striking proceeding works have been proposed to reshape the point clouds to the dense maps. VeloFCN [24] firstly takes the 3D point cloud as a 2D image date by projecting the point clouds to the image plane. Extended with multi-channel representations from the reflection, the range, the forward distance, the side distance, and the height, LMNet [14] achieves the real-time inferences of the surrounding cars in three-dimensional bounding boxes.

However, measuring from the roadside views is different from the ones on-board. In practice, there are two main problems inside this question:Inconsistent distribution. Mounted on overpasses or roadside poles with an oblique perspective, the LiDAR has a seriously inconsistent data distribution while the most effective points are located at a medium distance. This phenomenon is illustrated in the right picture of Figure 4.Roadblock interference. In the roadside scenes, billboards and metal signs are easy to be misjudged as small target vehicles by LiDAR points. It is also important for neural networks to reduce the noise of this input signal.

To alleviate the noises of roadblocks, we select the depth and height information as two input channels of our LiDAR subnet for it can effectively filtering out tremendous false positives. The horizontal values and the vertical values are neglected for the opposite reason. At the same time, we adopt prior reflection values to remove the road area for simplicity.

However, LiDAR cannot detect extremely large targets within the inconsistent data distribution and limited beams of the laser. To compensate for this shortage, the hue information in color space, which also shows insensitiveness to light intensity, is used as a supplementary input channel for our LiDAR subnet.

The overall data process includes three key steps. Specifically, we synchronize the collected point clouds with the image by projecting the points onto the image plane. These sparse projected point maps are further interpolated to generate dense maps. At last, we extract the hue information from the images and concatenate the data with original dense maps to form the inputs of the LiDAR subnet. The detailed steps are formalized as follows:Sparse LiDAR image generation: After space calibration between two kinds of sensors was finished, the matrix of camera extrinsic parameters $A_{4\times 4}$ and internal parameters $H_{3\times 4}$ are obtained. We further project a three-dimensional point P to the image plane by the following equation:
(6)p=sHAP
where *p* is the projected vector and *s* is a scale factor.Interpolation operation: Delaunay triangulation (DT) algorithm [25] and Voronoi diagram are introduced to fill in the sparse depth-map and height-map. DT can effectively interpolate all locations in the map even original position is far from the nearest point.Hue information extraction: we initially transform the color image from RGB space to HSL space, then stack the hue information together with depth and height map as the inputs. The hue information is extracted by the following equation:
(7)$$\bar{h}=\{\begin{array}{cc} 0^{{}^{\circ}}, & |if\;max\left(r,g,b\right)=min\left(r,g,b\right)\\ 60^{{}^{\circ}}\times \frac{g-b}{max\left(r,g,b\right)-min\left(r,g,b\right)}, & |if\;max\left(r,g,b\right)=r\;and\;g>b\\ 60^{{}^{\circ}}\times \frac{g-b}{max\left(r,g,b\right)-min\left(r,g,b\right)}+360^{{}^{\circ}}, & |if\;max\left(r,g,b\right)=r\;and\;g<b\\ 60^{{}^{\circ}}\times \frac{b-r}{max\left(r,g,b\right)-min\left(r,g,b\right)}+120^{{}^{\circ}}, & |if\;max\left(r,g,b\right)=g\\ 60^{{}^{\circ}}\times \frac{r-g}{max\left(r,g,b\right)-min\left(r,g,b\right)}+240^{{}^{\circ}}, & |if\;max\left(r,g,b\right)=b\; \end{array}$$
where $\bar{h}$, r, g, and b are the normalized hue information, the pixel value of the red, green, and blue element in each position, respectively.


After the above data processing steps, the inputs of the overall system are produced. The height, the depth, and the hue information are stacked and regularized as normal RGB images in the LiDAR subnet. Two cases of the inputs of both subnets are illustrated in Figure 5.

### 3.4. MME-YOLO: Multi-Sensor Multi-Level Composite Backbone

Using a cascading structure to improve the accuracy, CBNet [26] is a recent state-of-art that concentrates on using several same visual backbone networks to get robust feature maps. Not like the model ensemble system, CBNet achieves better performance by iteratively feeding high-layer features of assistant visual backbones into lower ones of the main visual backbone. However, this method has two drawbacks. First, the performance boost is sharply reduced when multi-subnet cascaded, which is shown in CBNet’s paper. In our approach, the multi-layer cascade idea of CBNet is adopted but the original redundant visual subnet is replaced by the meaningful LiDAR one. Second, the CBNet’s backbone modal tends to have more parameters for the consideration of better performance. Our detector adopts lightweight YOLOv3 as the backbone modal. The experiment shows that with the help of a multi-sensor multi-level composite structure, even the light model could achieve impressive improvements than traditional huge visual models.

The multi-sensor multi-layer YOLO cascade architecture (MME-YOLO) includes two subnets: the LiDAR subnet (denoted as L) and the Color image subnet (denoted as C). The original backbones in each subnet are inherited from YOLOv3. As seen in Figure 6, each subnet includes three levels, and each level consists of several convolutional layers with feature maps of the same size. Inspiring by CBNet architecture, we employ the feature maps of L to enhance the feature maps of C, by iteratively feeding the output features of L as part of input features to C in corresponding levels. Specifically, the input of the $l_{th}$ level of the subnet C (denoted as $\;x_{C}^{l}$) is the summary over the output of the previous ${\left(l-1\right)}_{th}$ level of C (denoted as $\;x_{C}^{l-1}$) and the output of the parallel level of the previous backbone L (denoted as $\;x_{L}^{l}$). This operation can be formulated as the following:(8)$$x_{C}^{l}=f^{l}\left(x_{L}^{l}\right)\;+\;g^{l}\left(x_{C}^{l-1}\right),\;2\geq l\geq 1$$
where $f^{l}\left(\cdot \right)$ denotes the composite lateral connection, which consists of a 1×1 convolutional layer to reduce the channels, batch normalization layer, and a bilinear upsample operation. Apart from this, the non-linear transformation $g^{l}\left(\cdot \right)$ is conducted in $l_{th}$ level of each subnet. An additional 3×3 convolution layer in a stride of two is implemented between different levels to reduce the resolution of feature maps.

Benefitting from the extraordinary centimeter-level resolution of the depth and height information, the MME-YOLO can significantly improve the identification ability when the visual subnet is broken down in dazzling situations, which is showed in the experiments part.

## 4. Experiments and Results

Our experiments consist of two parts. Generally, the performance of the proposed E-YOLO is examined and compared with several decent works on accessible traffic datasets LSVH [27] and UA-DETRAC [28], which respectively contains 14K and 82K pictures with various perspectives and vehicle densities in traffic surveillance. Moreover, to certify the realistic working performance of the E-YOLO in the cooperative roadside system, a minor 1.2 K visual dataset was collected at the exits of a highway. Comparison between E-YOLO and YOLOv4 [29] are illustrated in this visual dataset.

The other experiment was designed to test the proposed MME-YOLO in urban environments, including the sections during the daytime, the evening rush, and the night. In order to reduce the collision events in the intersection area, the monitoring area of our equipment is mainly the sections near the intersection, thus the cooperative system could have enough time for transmitting the warning signal to the other devices. The data frames of these scenarios are nearly 6 K. These dates were collected by a HIKVISION DX-2XM6000 camera and a Velodyne VLP-32 LiDAR. The focal length of the lens of the camera is 4mm and the range of the LiDAR is 200 m. The horizontal Angular resolution of the LiDAR adopted in our test is 0.1° and the running mode is the Dual Return. The sampled data are synchronized at 10 Hz through a multi-thread programming mechanism. The installation height is not fixed, ranging from 4.5 m to 6 m.

### 4.1. Implementation Details

Two related steps are utilized to training our network for better performance in all experiments. In the first step, we adopt the original weight file of YOLOv3 as the base stone, which is trained on the COCO2017 [30] trainval dataset. Besides, we use transferring learning to form a robust backbone structure for vehicle detection. In the second step, the above backbone and Kaiming initialized additional structure are trained together for 300 epochs with a weight decay of 0.0001 and a momentum of 0.9. A staircase-learning-rate schedule is applied with a 0.01 learning rate at the beginning. We multiple the learning rate with 0.5 in the 50, 100, 200 epochs, respectively. We use a mini-batch of size 32 for transferring learning and 16 for the overall proposed network. Additional data augmentation policies are random size crop, random horizontal flip, color jittering. All works are trained with an Intel I7-8700K CPU and two Titan Xp GPUs.

### 4.2. Experiments for the Visual Enhancement

To verify the effectiveness of our proposed E-YOLO, the experiments on the open dataset and the real working situations were both considered in this part. We first analyze the performance of the visual enhancement in the open dataset, then conduct the ablations studies on the structure of our visual model, and finally illustrate the detected results along with that of the YOLOv4-M in the realistic scenarios where the cooperative systems works.

The mean average precision of different models on open dataset LSVH and UA-DETRAC are compared in Table 1. Integrated with cross-level attention structure and ensemble inference head, E-YOLO gets a 72.48% mAP on LSVH and a 92.99% one on UA-DETRAC. Especially, compared with medium-scale and large-scale vehicles, the small ones have a relatively higher improvement on LSVH. This phenomenon shows that the proposed structures alleviate the scale imbalanced problem by a certain point. Figure 7 shows some detection results on LSVH and UA-DETRAC, respectively.

Since recent works are evaluated on several different GPU architectures for execute-time, we only operate several well-known works on the Pascal architecture and compare them with our proposed methods. Table 2 lists the mAP and the execute-time results of these projects inferencing on the 1080 Ti GPU. Our works achieve better performance compared with recent robust detectors, such as [31] and [32], who also combine multi-level features to perform vehicle detection.

To clarify the importance of designed structures, we construct several models by sequentially inserting the proposed attention structure and inference head based on YOLOv3. For example, we define A-YOLOtb as YOLOv3 with an additional top-down attention structure and anchor-based inference head. Alternatively, A-YOLObb, A-YOLOcb, and A-YOLOcf are referring to the bottom-up attention with the anchor-based head, the cross-level attention with the anchor-based head, and the cross-level attention with the anchor-free head, respectively.

As listed in Table 1, after the top-down attention structure was introduced, the overall detection system achieves satisfactory mAP improvement (2.47%) on LSVH, especially for the small vehicles (5.36%). On the contrary, the bottom-up attention plugin introduces more noises for small-scale vehicles (−1.63%), though the mean average precision of multi-scale vehicles boosted to 66.90%. This phenomenon certifies that abundant background cues bring detrimental effects to the small-scales but encouraging ones to the medium or the large ones. This also confirms that the balanced weight of background cues and semantical information is crucial to multi-scale detectors.

On top of the conventional anchor-based head, we also implied the anchor-free branch to strengthen the generalization of models in different scenes. After replacing A-YOLO_cb_ with A-YOLO_cf_, we observed that the mAP is further improved by 2.05% on UA-DETRAC in Table 1, which makes clear that the misallocated training labels and the heuristic anchor size design, generally described in anchor-free papers, contribute harmful training supervised signal for the neural network by a certain percent. Compared with A-YOLO_cb_ and A-YOLO_cf_ on both two datasets, the ensemble inference head achieves better detection ability than individual ones in all scales.

To better understand the effects of the attention structure, we illustrated the activation of the objectiveness in the small scales by upsampled the feature maps to unified resolution (640×384 pixels), which can be seen in Figure 8. In all three scenes, the activation of E-YOLO on the small-scale level shows an impressive improvement than YOLOv3. These attention pictures confirm that the cross-level attention guided feature selection blocks and ensemble inference head could equip the detector with stronger abilities toward the multi-scale problem.

Additional experiments are conducted in the collected highway data. The E-YOLO and the YOLOv4-M (the Median size one) were trained in this collected dataset where the cooperative system usually works. The qualitative images are presented in Figure 9. Although the mAP performance is elevated from 95.77% in E-YOLO to 96.13% in YOLOv4-M, the recall in Cars of the E-YOLO is slightly better than that of YOLOv4-M by 0.31%. Both of these two models achieve comparable performance in the highway scenarios.

### 4.3. Experiments for the Complex Scenarios in Urban Sections

To certify the validity of multi-sensor multi-level enhanced YOLO, more experiments are conducted on the collected Road-side Dataset, which includes both the LiDAR points and the images for nearly 6 K frames. As illustrated in Table 3, after LiDAR information was induced on several levels, the overall mAP performance of MME-YOLO is significantly improved compared with YOLOv3 by a large margin of 7.46%. Compared with the best-performed YOLOv4-L (the YOLOv4 model of largest size), the MME-YOLO also achieves a non-negligible advantage by 1.66%. When investigating the differences between E-YOLO and MME-YOLO, the relative mAP improvement of the Cars is boosted from 0.80% to 7.37% even under variable lighting conditions along with the occlusion, which means that the accurate distance and height information could make an extraordinary gain on the recognition of Cars. Additionally, several typical detection examples are illustrated in Figure 10 for a better understanding of the ability of MME-YOLO toward multi-scale vehicle detection.

To clarify the contribution of the LiDAR data, we followed the idea of CBNet and constructed the CB-YOLO_2 by two visual subnets. Except for the visual assistant backbone, all additional inference head and main backbone of MME-YOLO are kept. The results in Table 3 show that the MME-YOLO achieves more reliable performance than visual works especially for the Car category, which gained a relatively 4.75% improvement than CB-YOLO_2. Inspired by the LMNet, we accordingly stack the LiDAR maps and the images from the inputs of the convolutional network and construct the ME-YOLO (multi-sensor enhanced YOLO). Although inferencing from six input channels, ME-YOLO fails to catch up with the performance of MME-YOLO by a large margin, indicating that the recursive composite structure is better than fusion from the beginning. Additional experiments with YOLOv4 are also conducted in our experiments. However, test time augmentation (TTA) is not used in our experiments, which follows consistent settings with MME-YOLO. Compared with YOLOv4-L, MME-YOLO has an obvious advantage over most categories but falls behind in Van with fewer samples.

To quantify the differences between YOLOv4-L and MME-YOLO, an error analysis is conducted by counting the missed targets and wrong-positioning detections in results. As illustrated in the histogram in Figure 10, the errors are extensively reduced by MME-YOLO in all scenarios. Thanks to the benefits of the LiDAR signal, the missed targets decreased from 120 to 36 under backlighting conditions at night. Even in the well-illuminated daytime, the point information could help the visual detector to correct the misjudgments about the ambiguous backgrounds by eliminating wrong-positioning detections from 39 to 17.

Apart from regular experiments, we also evaluate the robustness of detectors along with the time. We note that MME-YOLO performs better than the visual E-YOLO and the original YOLOv4-L (without TTA) from Table 4 in abnormal lighting conditions. This again shows that the basic idea of multi-sensor and multi-level fusion strategy is a more effective dimension than unified visual works. As illustrated in Figure 10, while the baseline occasionally predicts a false negative or false positive result, our work keeps a robust detection ability under adverse lighting conditions.

Although MME-YOLO shows robust detection results in urban scenes, it faces challenges from the effective distance of the LiDAR. The experiments on the range of the detector are also considered in our research. We randomly tested five scenarios in our dataset within the time durations by logging the mean maximum measurement distance in each scenario. As illustrated in Figure 10, the minimum–maximum measurement distance is 93 m. As for the cars far than that distance threshold, the proposed model can only occasionally capture representative information.

To solve these problems, our future work will attend to taking time series into our system while considers replacing the cost LiDAR with economical sensors. At the same time, more weather conditions and more complex situations have been collecting and labeling for future object detection and object tracking purposes. Besides, we will attend to covering other targets in transportation scenarios, including pedestrians, motorcycle riders, and bicycle riders.

## 5. Conclusions

In this paper, we propose a multi-sensor multi-level composite fusion network for robust multi-scale vehicle detection under variable lighting conditions. The major creative work of this paper can be summarized as the LiDAR-Image composite module and enhanced inference head. In the composite module, the original point information is deeply analyzed and integrated into the visual backbone network at different levels, which significantly improved the abilities of the detector under abnormal lighting conditions. In the enhanced inference head, the cross-level attention blocks and the anchor-free/anchor-based ensemble system are developed to equip the positive grids with robust semantic information while penetrating the negative ones at different levels, which extraordinary improved the performance of the detector for multi-scale vehicle detection. Experiments indicate that our proposed structures can achieve reliable and accurate vehicle detection on both the traditional visual dataset and the collected LiDAR-Image Road-side Dataset.

## Figures and Tables

**Figure 1 sensors-21-00027-f001:**
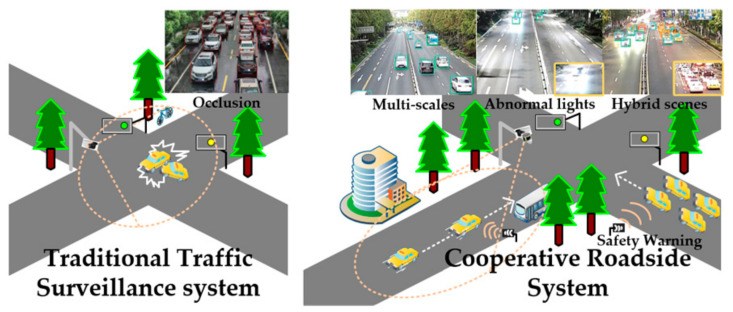
The differences between the traditional traffic management devices and the cooperative roadside system. While the traditional algorithms in traffic management devices concentrate on the occlusion scenarios, the ones in cooperative systems pay attention to the multi-scales and the robust detection ability towards the adverse environments.

**Figure 2 sensors-21-00027-f002:**
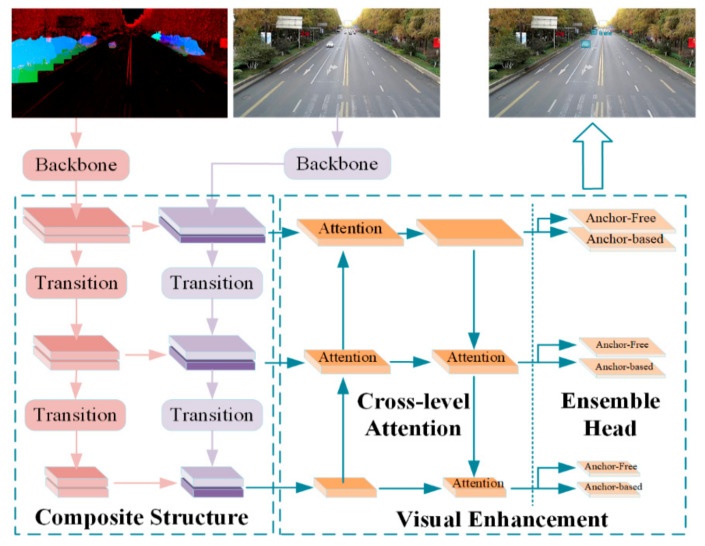
The overall design of the multi-sensor multi-level enhanced convolutional network architecture (MME-YOLO) network. Our work creatively transmits the feature maps of the LiDAR subnet to the Image subnet in a recursive manner. Additionally, we develop cross-level attention blocks to strengthen the merged features. Then, the ensemble head is introduced for better generalization in real scenes.

**Figure 3 sensors-21-00027-f003:**
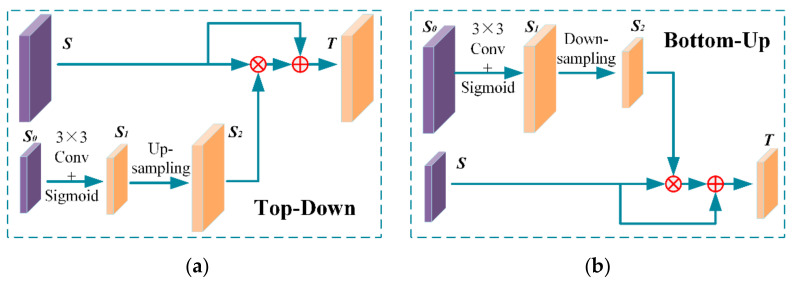
The attention guided feature selection blocks in cross-level attention module. (**a**) The top-down attention block; (**b**) The bottom-up attention block. Feature maps with the same size are multiplied or added in element-wise. These special designs of the attention blocks have greatly improved the performance of detectors, especially for small-scale vehicles. The capital T means the target feature maps and the capital S means the source feature maps. The symbol $S_{0},\;S_{1},\;S_{2}$ refer to the feature map from the upper or lower layer, the attention mask, and the attention mask with the same size as the source feature map, respectively.

**Figure 4 sensors-21-00027-f004:**
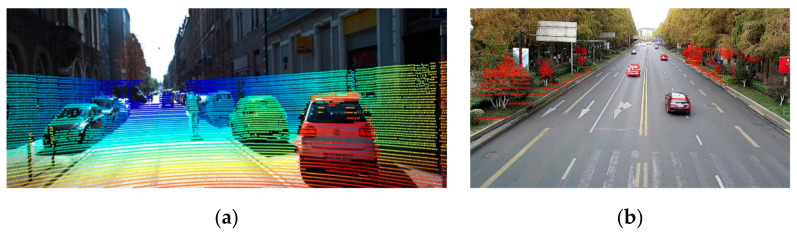
Examples of projected points on KITTI and private Roadside Dataset. We randomly choose an example in both datasets and project the points into the image. The outcomes are listed as the following: (**a**) the projected image on KITTI. These points are covered nearly all cars except for the occluded ones or far-off ones. (**b**) The projected image on Roadside Dataset. The distribution of unbalanced points makes it difficult to identify multi-scale vehicles. For example, the bottom right black car receives fewer points than the farther white one. We remove the points on the road for better visualization.

**Figure 5 sensors-21-00027-f005:**
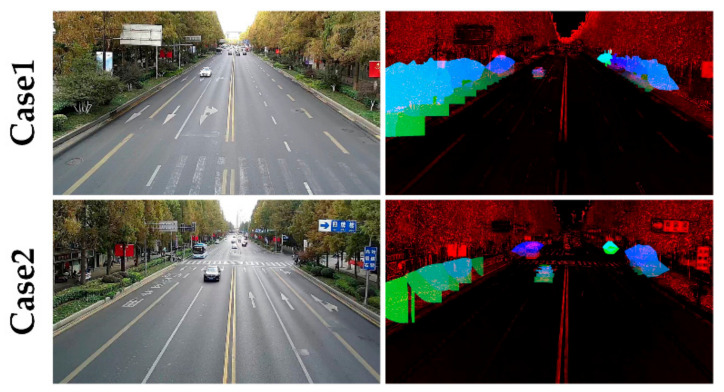
Two cases of the inputs of the Image subnet (the **left** column) and the LiDAR subnet (the **right** column). Case 1 shows a small-scale white car. Case 2 exhibits a light-blue bus and a medium-scale black car. From the visualizations of the LiDAR subnet input, the accurate distance and height value significantly contributes to the recognition of vehicles at semantic levels. However, the LiDAR-image also introduces inevitable noises into low-level cues, such as the edges. Therefore, it is acceptable that fusion from the semantic levels is more useful than fusion from the beginning level.

**Figure 6 sensors-21-00027-f006:**
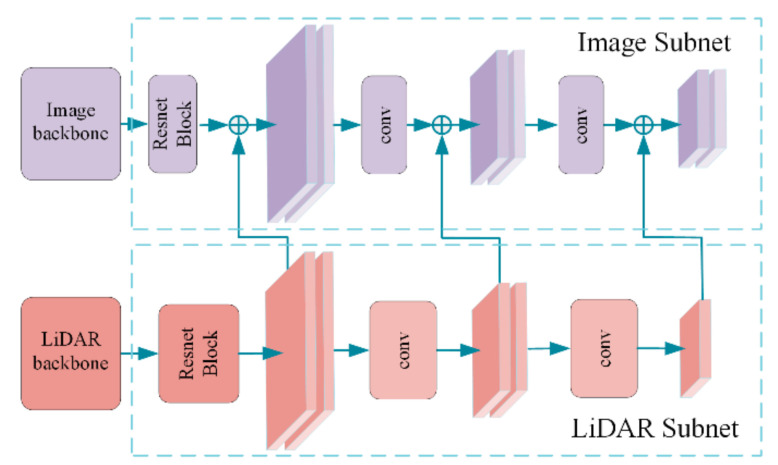
Visualization of the multi-sensor multi-level composite module. The feature maps in the LiDAR subnet are added to the feature maps in the Image subnet in a recursive manner. This idea was originally published by CBNet [26]. We extend the composite methodology into the multi-sensor fusion field with a recursive multi-sensor multi-level composite structure.

**Figure 7 sensors-21-00027-f007:**
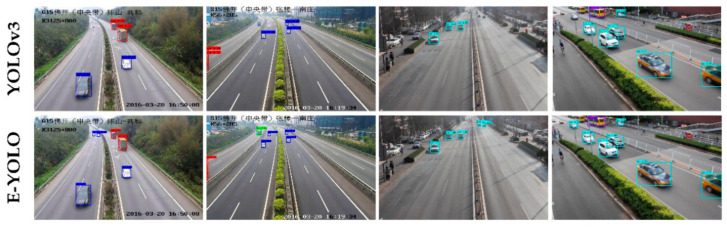
Detection Examples of YOLOv3 and E-YOLO (the visual enhanced YOLO) on LVSH (columns 1 and 2) and UA-DETRAC (columns 3 and 4). After plugged with the proposed attention structure and ensemble inference head, E-YOLO can easily detect the small-scale vehicles that are timely neglected by YOLOv3. The Chinese characters in the figures are the locations of the cameras, which are not used in our experiments. Best viewed on the screen.

**Figure 8 sensors-21-00027-f008:**
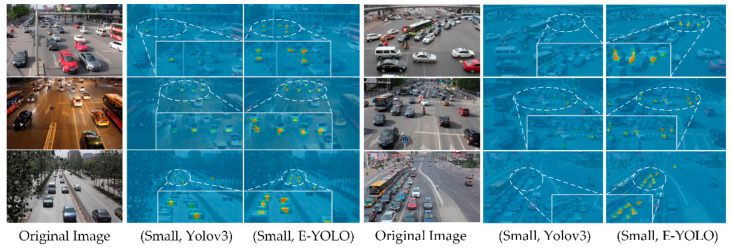
The activation ranks of the objectiveness of two models in the small-scale vehicles. Pictures in the second column of each scene are the activation ranks of traditional YOLOv3, and the third column is the ones of E-YOLO. Objects are identified with different colors in different objectiveness ranks. The colors from light blue to orange-red are defined as the confidence from zero to one by indexing the confidence rank in HTML Color Table. Best viewed on screen.

**Figure 9 sensors-21-00027-f009:**
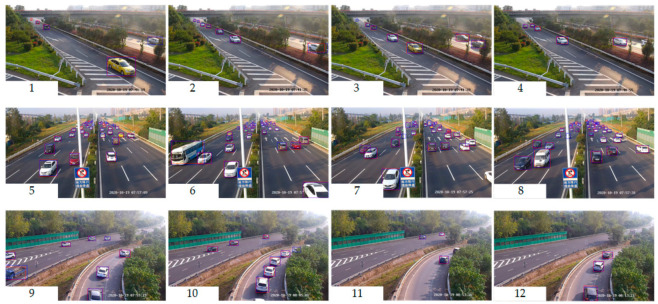
Qualitative detections of the E-YOLO and the YOLOv4-M on highway scenarios. The red bounding boxes are predicted by E-YOLO, the Blue bounding boxes are predicted by YOLOv4-M. Compared with the YOLOv4-M, which has a better detection toward the occlusion, the E-YOLO shows excellent performance on multi-scale recognition, especially for the cars. Best viewed on screen.

**Figure 10 sensors-21-00027-f010:**
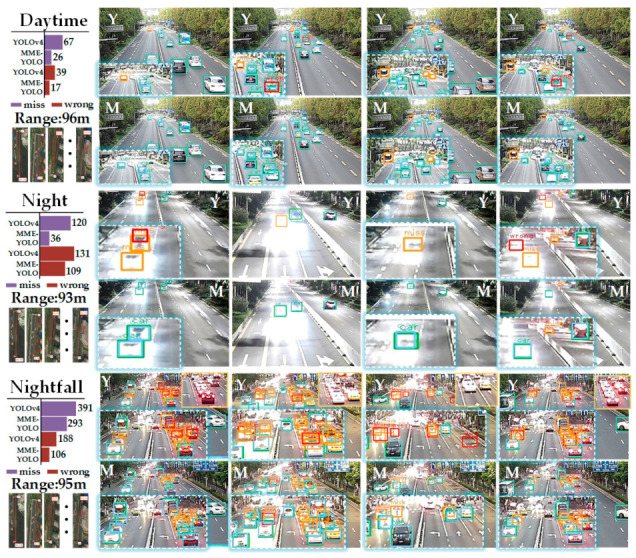
Comparison between YOLOv4-L and MME-YOLO on the Road-side Dataset. Error histograms showed in the top-left corner. Minimum–maximum distances are listed in the bottom-left corner. Qualitative images are shown on the right. Light blue bounding boxes refer to the ground truth; red ones mean that wrong positions were predicted; orange ones mean that missed targets; green ones mean matched detections. We manually enlarged the areas for better visualization.

**Table 1 sensors-21-00027-t001:** The average precision of visual detectors on LVSH and UA-DETRAC. We split the original test set into the small-scale set, the medium-scale set, and the large-scale set for a fair comparison of different models on multi-scale inference. For example, AP_S_ refers to the average precision on the small-scale set.

Model	mAP (%) ^1^	AP_S_	AP_M_	AP_L_	mAP ^2^	AP_S_	AP_M_	AP_L_
YOLOv3	66.65	61.87	70.34	67.73	88.11	81.23	93.73	89.37
E-YOLO	72.48	70.79	72.73	73.91	92.99	89.03	96.11	93.83
A-YOLO_tb_	69.12	67.23	72.39	67.75	90.41	85.35	94.21	91.68
A-YOLO_bb_	66.90	60.24	71.51	68.94	89.06	80.91	94.23	92.03
A-YOLO_cb_	69.96	66.34	71.43	72.12	90.79	85.81	94.39	92.17
A-YOLO_cf_	71.95	69.71	72.71	73.43	92.84	88.29	96.42	93.81

^1^ The Performance of models on LVSH. ^2^ The Performance of models on UA-DETRAC.

**Table 2 sensors-21-00027-t002:** Comparison of models on UA-DETRAC. We copied the released data on the website of UA-DETRAC and the results in several recent papers. To quantitate the improvement of our work, experiments on same 1080 Ti GPU with recent works are conducted. The caption of Multi-level means the usage of the combination of multi-level features inside each model.

Model	Input Size	Multi-Level	FPS	mAP (%)
DPM [33]	640×480	False	0.23 (I7-8700 K)	25.71
ACF [34]	640×480	False	0.81 (I7-8700 K)	46.37
Faster R-CNN(VGG16) [35]	-	False	11.23	72.67
SSD300 [36]	300×300	False	58.78	74.18
SSD512 [36]	512×512	False	27.75	76.83
RefineDet320 [37]	320×320	False	46.83	76.97
RefineDet512 [37]	512×512	False	29.45	77.68
DP-SSD300 [31]	300×300	True	50.47	75.43
DP-SSD512 [31]	512×512	True	25.12	77.94
CMNet [32]	416×416	True	47.49	91.71
YOLOv3 [1]	416×416	True	51.26	88.11
A-YOLO_cf_	416×416	True	38.87	92.84
E-YOLO	416×416	True	28.89	92.99

**Table 3 sensors-21-00027-t003:** Comparison of models on the private Road-side Dataset. We intensively compare the proposed model with the baselines in different categories for AP and Recall. The multi-sensor multi-level composite module improves the performance of visual detectors impressively.

Model	mAP(%)	AP (%)			Recall (IOU >0.50) (%)		
		Car	Bus	Van	Car	Bus	Van
YOLOv3	84.17	84.51	92.43	75.58	86.52	92.89	78.13
E-YOLO	88.59	85.31	94.93	85.53	88.47	96.12	86.38
MME-YOLO	91.63	91.88	97.60	85.41	93.21	97.14	87.23
CB-YOLO_2	88.19	87.13	95.02	82.42	88.11	95.57	81.30
ME-YOLO	84.31	84.59	94.13	74.21	87.86	92.13	78.06
YOLOv4-S	84.21	85.23	92.36	75.03	80.23	93.24	83.22
YOLOv4-M	88.63	85.36	92.56	87.98	85.90	97.30	87.33
YOLOv4-L	89.97	86.93	93.55	89.44	86.01	97.32	87.40

**Table 4 sensors-21-00027-t004:** Comparison of models on the Road-side Dataset under different lighting conditions. The MME-YOLO is especially suitable for the backlighting conditions at night. Although the low-level edges were overwhelmed by turbulence lighting, the detector can accurately recognize the vehicles on the road.

Model	Average (%)	Daytime	Night	Nightfall
YOLOv4-L	89.97	98.13	87.23	84.55
E-YOLO	88.32	97.37	83.84	83.75
MME-YOLO	91.18	98.21	89.03	86.31

## Data Availability

The data presented in this study are available on request from the corresponding author. The data are not publicly available now due to deficient maintenance capacity.

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
