# Peer review of "MME-YOLO: Multi-Sensor Multi-Level Enhanced YOLO for Robust Vehicle Detection in Traffic Surveillance"

_sensors, 2020, doi:10.3390/s21010027_

Round 1

Reviewer 1 Report

This paper proposes a novel multi-sensor multi-level enhanced convolutional network architecture, called MME-YOLO, with consideration of extremely imbalanced vehicle scales and variable lighting conditions. MME-YOLO consists of two tightly coupled structures, i.e., the enhanced inference head and the LiDAR-Image composite module. More specifically, the enhanced inference head preliminarily equips the network with stronger inference abilities for redundant visual cues by attention-guided feature selection blocks and anchor-based/anchor-free ensemble head. Furthermore, the LiDAR-Image composite module cascades the multi-level feature maps from the LiDAR subnet to the image subnet, which greatly strengthens the generalization of the detector under abnormal lighting conditions. The enhanced inference head achieved 5.83% and 4.88% mAP improvement then YOLOv3 on visual dataset LVSH and UA-DETRAC, respectively. Integrated with the composite module, the overall architecture gains 91.18% mAP in the collected Road-side Dataset. My questions on this paper are summarized as below:

  • The proposed design increases the vehicle detection accuracy by about 5% mAP improvement as compared to YOLOv3 with the help of Lidar and camera fusion. Please clarify the lidar spec adopted in the experiments.
  • YOLOv3 is not the state-of-the-art object detection model. I would suggest the authors to compare the proposed design with more latest models like YOLOv4 or CenterNet.
  • For the application of Road Side Unit, there are other moving objects in addition to vehicles, like motorcycle riders, bicycle riders, as well as pedestrian. How is the proposed design applied to detect moving objects other than vehicles?
  • For the performance comparison in Table 2, please specify the spec. of the adopted GPU in running the model.
  • Please evaluate the max. detected distance of vehicles in the proposed design.
  • In Table 4, the detection accuracy of the proposed design is decreased under low illumination case (from 98.76% mAP to 76.65% mAP). However, the proposed design adopts the fusion of lidar and camera together to improve the vehicle detection accuracy, while lidar is very robust under low illumination situation. Please evaluate the potential reasons and try to see if there are remedies to deal with the low illumination case.

Author Response

Dear Reviewers:

Thanks a lot for your precious time and thorough review of the manuscript entitled “MME-YOLO: Multi-sensor Multi-level Enhanced YOLO for Robust Vehicle Detection in Traffic Surveillance” (Manuscript ID: sensors-976393). Here, we would like to express our sincere appreciation to all of you for the valuable comments, which are very important to improve the quality of the paper.

According to the comments and suggestions from the editors and reviewers, we have revised the manuscript carefully. The comments and our responses are listed in detail in the attachment. We use orange text for our responses, and all changes are marked as blue in the revised manuscript.

Finally, the authors would like to express our gratitude again to the editors and reviewers for the valuable comments and suggestions, as well as the time and efforts spent in the review.

With the insightful comments and suggestions made by the editors and reviewers, we are able to enhance and improve the quality of the paper.

If there are other problems or further requirements, please contact us in time.

Sincerely yours,

The authors.

Reviewer 2 Report

This paper proposes Multi-sensor Multi-level Enhanced YOLO for Robust Vehicle Detection in Traffic Surveillance. As compared to yolov3, the latest version YOLOv4 shows better accuracy. I suggest the author should compare their findings with yolov4 and YOLOv4-tiny.

Author Response

(The authors gave the same response as above.)

Round 2

Reviewer 1 Report

The authors have addressed most of my review comments and the experimental results show that the proposed design has good performance compared to the existing methods. Therefore, I suggest this paper could be published in the current form.